# Durability of Steel Fiber-Reinforced Concrete Containing SiO_2_ Nano-Particles

**DOI:** 10.3390/ma12132184

**Published:** 2019-07-07

**Authors:** Peng Zhang, Qingfu Li, Yuanzhao Chen, Yan Shi, Yi-Feng Ling

**Affiliations:** 1School of Water Conservancy and Environment, Zhengzhou University, Zhengzhou 450001, China; 2Changjiang River Scientific Research Institute of Changjiang Water Resources Commission, Wuhan 430010, China; 3Departmen of Civil, Construction and Environmental Engineering, Iowa State University, Ames, IA 50011, USA

**Keywords:** steel fiber, concrete, durability, nano-particle

## Abstract

An experimental study was conducted to investigate the effect ofnano-SiO_2_ and steel fiber content on the durability of concrete. Five different dosages of nano-SiO_2_ particles and five volume dosages of steel fiber were used. The durability of concretes includes permeability resistance, cracking resistance, carbonation resistance, and freezing-thawing resistance, and these were evaluated by the water permeation depth, number of cracks, total cracking area per unit area of the specimens, carbonation depth of the specimens, and the relative dynamic elastic modulus of the specimens after freezing-thawing cycles, respectively. The results indicate that the addition of nano-SiO_2_ particles significantly improves the durability of concrete when the content of nano-SiO_2_ is limited within a certain range. With the increase of nano-SiO_2_ content, the durability of concrete first increases and then decreases. An excessive number of nano-SiO_2_ particles could have an adverse effect on the durability of the concrete. The addition of the correct amount of steel fibers improves the carbonation resistance of concrete containing nano-particles, but excessive steel fiber reduces the carbonation resistance. Moreover, the addition of steel fibers reduces the permeability resistance of concrete containing nano-particles. The incorporation of steel fiber enhanced the freezing-thawing resistance and cracking resistance of concrete containing nano-particles. With increasing steel fiber content, the freezing-thawing resistance of the concrete containing nano-particles increases, and the cracking resistance of the concrete decreases gradually.

## 1. Introduction

Concrete has been widely used in structural engineering because of its superior characteristics, such as simple preparation techniques, low energy consumption, high durability, low price, and good bonding performance with reinforcing steel bars and plates. However, there are also a number of disadvantages when using concrete materials, including high weight, low toughness, and low tensile properties, which restrict their widespread application. One important method for modifying concrete is incorporating fiber materials into concrete composite, which can significantly enhance the mechanical performance, deformation behavior, and anti-cracking performance [1]. Typically, there are numerous types of fibers suitable to be used in concretes, such as polypropylene fiber [2], glass fiber [3], plant fiber [4], carbon fiber [5], polyvinyl alcohol fiber [6], basalt fiber [7], and steel fiber [8]. Of these fibers, steel fibers are the most commonly applied in concrete structures. Over the past two decades, numerous researchers have studied the mechanical properties, durability, physical properties, and microstructural performance of concrete reinforced by steel fibers. Pajak and Ponikiewski reported the impact of steel fiber shape, at similar slenderness ratios, on the mechanical behavior of steel fibers reinforced self-compacting concrete [9]. Ma et al. reported the mechanical performance of autoclaved lightweight concrete made of shell-aggregate reinforced by steel fibers, and compared the properties of concrete composite made of crushed stones and sintered concrete. Their results indicated that the use of steel fibers in autoclaved lightweight shell-aggregate concrete significantly improved the flexural strength of concrete [10]. Mo et al. reported the toughness, bond performance, and durability of lightweight concrete reinforced by steel fibers, which was manufactured using oil palm shell reinforcement. The results indicated that a small amount of steel fibers resulted in no degradation in the permeation properties, whereas higher steel fiber dosages would lead to clear growth in the water absorption and sorptivity of the concrete [11]. Afroughsabet reported that the incorporation of steel fibers significantly improved the mechanical properties of recycled aggregate concrete [12]. Zhang et al. reported that steel fiber had a significant influence on fracture behavior of concrete composites containing nano-particles, and the fracture toughness and fracture energy increased gradually when steel fiber fraction increased from 0.5% to 2% [13].

Although the durability of concrete composite could be enhanced by the incorporation of steel fibers, it remains necessary for the concrete used in severe and highly corrosive environments to enhance its durability. In particular, after a number of cracks appear, the durability of concrete structures is significantly decreased because of disease caused by the cracks [14,15,16]. With the development of nanotechnology, there are numerous potential application of nano-materials in concrete materials. Nano-materials exhibit the unique nanometer effect because nano-particles are miniscule, and have large specific surface areas, interface energy, and a significant percentage of surface atoms [17]. Because of the above-mentioned nanometer effects, the addition of nano-particles can improve both the performance of the hardened cement paste and the bond performance between aggregates andcement pastes [18]. The incorporation of nano-particles to concrete has attracted increasing interest of numerous researchers, and a significant number of studies have been conducted over the past decadeon the performance of concrete incorporating nano-particles. Zhang and Li reported the abrasion resistance, flexural fatigue performance, porous structure, and chloride ion penetration of pavement concrete incorporating nano-SiO_2_ and nano-TiO_2_, and their results indicated that the incorporation of nano-SiO_2_ and nano-TiO_2_ significantly enhanced the durability and fatigue properties of concrete [19,20,21]. Salemi and Behfarnia reported that the application of nano-SiO_2_ and nano-Al_2_O_3_ together with polypropylene fibers can enhance freeze-thaw resistance and compressive strength of pavement concrete because of the reduced penetration and porosity of concrete [22]. Ismael et al. investigated the impact of Al_2_O_3_ and SiO_2_ nano-particles on bonding properties between steel and concrete, and reported that the addition of nano-particles resulted in an enhancement in bonding strength on concrete with higher cement dosages [23]. Wu et al. reported that the flexural properties and fiber-matrix bonding performance of ultra-high performance concrete could be significantly improved by the incorporation of 3.2% nano-CaCO_3_ [24].

These nano-particles, together with silver nano-particles, carbon nano-tubes, and nano-fibers, are typically used for the production of commercially available building products containing nano-objects [25]. To date, numerous types of nano-particles have been applied to improve the properties of concrete, including nano-metakaolin [6], nano-fly ash [26], nano-limestone [27], nano-TiO_2_ [28], nano-Fe_3_O_4_ [29], nano-Al_2_O_3_ [23], nano-CaCO_3_ [30], and nano-SiO_2_ [31]. Of these, nano-SiO_2_ is most commonly used in concrete. SiO_2_ nano-particles can accelerate generation of calcium silicate hydrate (C-S-H) gel and dissolution of tricalcium silicate (C_3_S) because of their hot activity, which is significantly affected by the size of SiO_2_, and nano-SiO_2_ can supply nucleating placefor the inclusion of C-S-H [32]. The durability of concrete composites has significant importance for the design of mix proportion and actual engineering application of concrete composites, particularly for structures under severe and highly corrosive environments. The durability and service life of concrete structures are dependent on the environmental conditions in which there are different types of aggressive materials. There is close link between two durability properties of concrete (permeability resistance, cracking resistance, carbonation resistance, and freeze-thaw resistance). For concrete structures, water is the primary cause of damage. In the presence of water, concrete damage such as water seepage, and freeze-thaw damage, could occur. The structure with low cracking resistance may have bad permeability resistance, carbonation resistance, and freeze-thaw resistance. If the concrete has low cracking resistance, this damage to structures will further deteriorate, which could in turn result in safety and reliability issues in engineering structures. Presently, the bulk of studies have considered the effects of SiO_2_ nano-particles on mechanical, and fracture properties, and bonding performance of steel fiber reinforced concrete [13,23]. However, only limited studies have conducted systematic investigations on the influence of nano-SiO_2_ on the durability of concrete composite reinforced by steel fibers. In this study, durability experiments (permeability resistance, cracking resistance, carbonation resistance, and freeze-thaw resistance tests) were conducted to determine the influence of nano-SiO_2_ and steel fibers on the durability of the concretes composite, which is the novelty of this study.The results of this study could provide design guidance for the application of concrete composite containing nano-SiO_2_ reinforced by steel fibers in actual engineering structures. In this study, the durability of steel fiber reinforced concrete containing nano-SiO_2_ is investigated.

## 2. Experimental Program

Systematic durability experiments (carbonation resistance test, permeability resistance test, freeze-thaw resistance test and cracking resistance test) were conducted to determine the influence of nano-SiO_2_ and steel fiber on the durability of concrete.

### 2.1. Materials

Chinese Class 42.5R Ordinary Portland cement (Mengdian Cement Factory in Xinxiang City of China) was used, and its primary performance parameters are presented in Table 1. Class I fly ash (obtained from coal-fired plants) was used, and the physical properties are presented in Table 1. Broken stones (with a maximum grain diameter of 26.5 mm) and fine aggregate (with a fineness modulus of 2.6) were also used. The milling steel fiber used in this study was produced by Yujian Steel Fiber Co., Ltd., Zhengzhou City, China, and its properties are presented in Table 1. The nano-SiO_2_ added in the experiments was in loose amorphous powder form, and the content of SiO_2_ was greater than 99%. Table 2 presents the properties of the nano-particles. The flowability of the fresh concrete was improved using a high-efficiency, polycarboxylate-water reducing admixture. The water-reducingratio of this reducing admixture was 14%. The raw materials used are shown in Figure 1. The dosages of the fly ash and nano-particles were determined by substituting the same amount ofcement, and steel fibers were added in composites by holding constant the amount of binding material. The quantity of fly ash by mass was 15%, and the quantity of nano-particles by mass varied from 1% to 9% (the weight percentage is calculated with respect to the total weight of the cement and fly ash). Referring to the common dosage ranges of nano-SiO_2_ and steel fiber used in concrete in other studies and our previous study [8,13,33], five different contents of nano-SiO_2_ particles (1%, 3%, 5%, 7%, and 9%) and five volume dosages of steel fiber (0.5%, 1.0%, 1.5%, 2.0%, and 2.5%) were used. The total binder content was controlled as 543.4 kg/m^3^. Table 3 presents the contents of binding materials and steel fiber used in this study. It is known from our previous study that the addition of 5% nano-SiO_2_ exhibits the greatest increase in the mechanical properties and fracture performance of concrete composites [8,13]. Therefore, in Mixes 7–11, the content of nano-SiO_2_ was unchanged at 5%.

### 2.2. Mixing of Fresh Concrete

Because of the miniscule size of nano-particles, it is difficult for nano-SiO_2_ to be dispersed uniformly in the concrete mixture. The nano-particles and steel fibers cluster if the mixing procedure of the concrete composite is not appropriate. In this experiment, a horizontal agitator was applied to prepare fresh concrete in order to prevent clustering of the nano-particles and steel fibers. The appropriate mixing program was determined by trial and error. The fine and coarse aggregates were mixed together for 30 s, and then the nano-materials, cement, and fly ash were poured into the mixture to be stirred for a further 30 s. The steel fibers were then added and mixed for 30 s. Finally, the water containing the water reducing agent was incorporated and mixed for a final 60 s. The distribution of the fibers and nano-particles clearly affect the workability of fresh concrete and the durability of hardened concrete. During the course of the mixing, the fresh concrete mixture was randomly sampled to observe the dispersion of the nano-materials and fibers. From the workability of the fresh concretes and fracture surface of the specimens, the uniform distribution of steel fibers and nano-particles could be determined.

### 2.3. Carbonation Resistance Test

Carbonation resistance test was carried out in the carbonation box (CCB-70A), which was manufactured by Suzhou Donghua Test Instrument Co., Ltd., Suzhou City, China. Figure 2 presents the apparatus of carbonation test of concrete composite. The carbonation box is a closed container with an air-tight door. In the box, there is a gas analyzer for CO_2_, and it can monitor the concentration of CO_2_, which should be 20%. The CO_2_ gas was supplied from the steel cylinder through a pipe to the carbonation box. The size of the cube carbonation specimen was 100 × 100 × 100 mm [34]. The depth of carbonation of the samples can be used to assess the carbonation resistance of concrete. Each group comprised three samples, and the average carbonation depth was taken to be the conclusive carbonation depth. The samples were cured to the curing period under standard conditions (temperature: 20 ± 2 °C and relative humidity greater than 95%) before the test. The specimens were then left in a drying oven at a temperature of 60 °C for 48 h. After drying, the surfaces of the samples, except for two symmetric side surfaces, were covered by a thin layer of molten paraffin. Before the specimen was placed into the carbonation box (as shown in Figure 2), a series of parallel lines were marked on the unsealed surfaces, using a color pen, to assist with measuring the carbonation depth. After the test finished, the specimen was split in half, and the fracture surface was sprayed with a phenolphthalein alcohol solution with a concentration of 1%. After about 30–60 s, the color of the carbonation part of the specimen changed. In addition, the carbonation depth of the specimen on each measuring point was measured with a steel ruler. Finally, the average carbonation depth of the specimen could successfully be measured.

### 2.4. Permeability Resistance Test

The dimensions of the circular truncated cone specimens for the permeability resistance test were Φ175 × 150 × Φ185 mm [35]. The permeability resistance test was conducted on a fully automatic permeability machine (HP-40), which was manufactured by Cangzhou Risheng Test Instrument Co., Ltd., Cangzhou City, China. Figure 3 shows the apparatus used in the concrete composite anti-permeability test. After drying, the specimen was sealed with a layer of paraffin. In addition, then the specimen was compacted into the heated mould through the machine of screw compressor. The specimen with the steel mould was then put on the fully automatic permeability instrument, and the mould was fixed using some high-strength bolts. The water permeation depth of the sample sunder water pressure could be used to assess the permeability resistance of concrete. Each group comprised six samples, and the average water permeation depth was taken as the conclusive water permeation depth. During the test, the specimens were subjected to pressurized water, at a pressure of 1.2 MPa, for 24 h. The pressure was then withdrawn, and the specimens were split in half. On the split surface of the specimen, dots were placed on each of 10 equidistant lines to indicate the water permeability depth (Figure 3b). The average water permeability depth of the 10 dots was taken as the water permeability depth of the specimen.

### 2.5. Freeze-Thaw Resistance Test

Freeze-thaw resistance of concrete can be evaluated by the quick-freezing method after a certain number of freeze-thaw cycles of specimens. The size of the prism freeze-thaw samples was 100 × 100 × 400 mm [36]. The specimens were placed at an ambient temperature for 24 h after they were cast. They were then cured to the test age under standard conditions (temperature: 20 ± 2°C and relative humidity greater than 95%) after demolding. The specimens were immersed in water at the temperature of 20 ± 2 °C for 4 days, after which their weight and dynamic moduli of elasticity were measured. The specimens were then placed in the freeze-thaw test machine (HC-HDK), which was manufactured by Jianyan Huace Instrument & Equipment Co., Ltd. Beijing City, China. According to the standard [36], the relative dynamic elastic modulus can be used as the evaluation parameter on freeze-thaw resistance of concrete after a certain number of freeze-thaw cycles. The relative dynamic elastic modulus of specimens was measured after each 25 cycles of freezing-thawing, and the relative dynamic elastic modulus could be obtained. In this study, relative dynamic elastic modulus of the specimen subject to 100 cycles of freeze-thaw was determined to evaluate the freeze-thaw resistance of the concrete composite. Figure 4 shows the apparatus used in the freeze-thaw cycle test and the dynamic elastic modulus measurement. The water surface in the specimen carrier should be 10–20 mm higher than the top surface of the specimen. Each freeze-thaw cycle should be completed within 2–4 h.

### 2.6. Cracking Resistance Test

The cracking resistance test was carried out according to the standard [36], and the size of the square block cracking resistance specimens were 600 × 600 × 63 mm. The specimens were restricted by a square steel angle mould, which was reinforced by stiffening ribs around the mold. The cracking resistance instrument was manufactured by Torrent Instrument Co., Ltd. Shanghai City, China. Each group comprised 2 specimens. Figure 5 shows the test setup. The specimens were cast under the constant certain temperature (20 *±* 2 °C) and relative humidity (60% *±* 5%). The fresh concrete composites were leveled using a trowel immediately after they were poured into the mould with the surface higher than the upper surface of the mould frame. The surface of the plate specimen should be smooth. Then, the specimens were covered using a piece of plastic cloth, which was taken away after the specimens were maintained at a temperature of 20 ± 1 °C for 2 h. The electric fan standing 150 mm away from the short edge of the mould was turned on immediately after the fresh concrete composite was cast in the mould. The wind speed on the transverse center line of the specimen should be controlled as 4–5 m/s when the electric fan was blowing to the specimen surface. An iodine-tungsten lamp with the power of 1000 W was fixed 1.2 m over the transverse center line of the specimen to shine the surface of the specimen. The electric fan and iodine-tungsten lamp were continuously working for 24 h and 4 h, respectively, before they were turned off. Then the widths of the cracks on the specimen surface were measured one by one using crack width measuring device. According to the measured crack widths, the crack lengths were measured hierarchically. During the test, the initial cracking time, crack length, crack width, and number of crack were recorded. The average cracking area of a single crack, the crack numbers per unit area of the specimen, and the total cracking area in unit area of the specimen can be calculated, respectively, as follows [36]:(1)$$a=\frac{1}{2N}\sum_{i}^{N}W_{i}\times L_{i}$$
(2)$$b=\frac{N}{A}$$
(3)C=a×b
where *a* is the average cracking area of a single crack, mm^2^; *N* is the total number of cracks; *W*_i_ is the maximum width of the ith crack, mm; *L*_i_ is the length of the ith crack, mm; *b* is the number of cracks per unit area of the specimen; *A* is the area of the specimen surface, 0.36 m^2^; and *C* is the total cracking area per unit area of the specimen, mm^2^/m^2^.

## 3. Results and Discussion

### 3.1. Carbonation Resistance

Figure 6 shows the variations in the carbonation depth of the specimens under different carbonation ages with increasing SiO_2_ nano-particles content. As can be seen, the carbonation depth of the sample increases with increasing carbonation age. The carbonation depth first decreases, but then increases incrementally with increasing nano-particle dosage for similar carbonation ages. The minimum carbonation depth is observed when the nano-particle content reaches 7%, where the depth is 22.7% smaller than that of basic concrete without nano-SiO_2_.

During the complex carbonation process, CO_2_ reacts not only with portlandite, but possibly also with other cement hydrates, primarily the calcium silicate hydrate (C-S-H) gels [37]. During the cement hydration reaction process, an amount of Ca(OH)_2_ is generated, and most of them have adverse effect to the mechanical properties and durability of concrete. There will be a second hydration reaction due to the addition of nano-SiO_2_. In the reaction, an amount of Ca(OH)_2_ inside the composite is consumed, and C-S-H gels are generated. Large amounts of C-S-H gel can increase the density of the concrete composite. There is an optimal content of nano-SiO_2_ for reinforcement in carbonation resistance of concrete. After the dosage of SiO_2_ nano-particles increases beyond the optimal content, the nano-particles are difficult to be dispersed uniformly. Therefore, the excessive nano-SiO_2_ content may have an adverse effect on the density of concrete, which results in an increase in carbonation depth of the sample. The influence of nano-SiO_2_ is similar to silica fume on carbonation resistance of concretes. Yang reported that the carbonation depth of the sample will decrease stepwise as the silica fume content increases from 0% to 5%, while the depth increases as the silica fume content increases from 5% to 10% [38]. The active SiO_2_ in silica fume can react with the cement hydration products, which decrease the alkalinity of the concrete. However, a significant amount of calcium silicate gel from the reaction can fill macropores inside the concrete, which could reduce the porosity of concrete and improve the strength and density [39].

The relationship between the carbonation depth of the specimens and the steel fiber dosage, containing 15%fly ash and 5% nano-SiO_2_, is shown in Figure 7. It can be seen that the carbonation depth of steel fiber-reinforced concrete containing nano-particles increases steadily when the carbonation age increases from 3 to 14 days, and a significant increase is observed when the age increases from 14 to 28 days. In addition, the incorporation of steel fibers has a significant impact on the carbonation depth of concrete containing nano-particles. With the steel fiber content increasing from 0% to 1.5%, the carbonation depth decreases gradually, and increases when the fiber fraction increases to 2.5%. The minimum carbonation depth is observed at a nano-particle content of 7%. Compared to the control concrete without steel fibers, the carbonation depth of the concrete specimen reinforced by 1.5% steel fiber decreases by 15.4%.

After the concrete specimen was cast, a significant number of small holes—which facilitate the diffusion of CO_2_ in concrete—were observed in the cement paste because of the evaporation of free water and chemical shrinkage. The incorporation of steel fiber can efficiently block the diffusion channel of CO_2_ and increase the resistance to CO_2_ diffusion. Consequently, this decreases the carbonation speed of concrete [40]. In other words, the effect of steel fibers on the carbonation speed of concrete can be considered as the reinforcement of steel fibers on the microstructure of concrete materials. Therefore, the reason why the steel fibers delayed the carbonation of concretes can be illustrated by the improvement of steel fiber on the microstructure of concrete. However, with further increases in the steel fiber dosage, a significant amount of steel fibers—distributed randomly in the concrete—began to prevent the cement paste from filling the macropores, which increased the internal porosity of the concrete. However, if the water–binder ratio remained unchanged, an excessive amount number of steel fibers would cluster in the concrete composite, which would result in defects inside concrete [41]. An excessive steel fiber fraction would result in numerous microcracks generated inside the concrete, and a significant number of microcracks can provide new channels for CO_2_ to penetrate the concrete. Therefore, an excessive steel fiber fraction would decrease the carbonation resistance of concrete.

### 3.2. Permeability Resistance

Permeability is a fundamental material property for characterizing concrete durability because it determines the penetration of aggressive substances responsible for degradation under a pressure gradient [42]. Concrete is a kind of highly porous media and its permeability resistance may be influenced by different measuring method [42,43]. In this study, water permeation depth was used to evaluate permeability resistance of concrete. The water permeation depths of 15% fly ash concrete specimens incorporating 0%, 1%, 3%, 5%, 7%, and 9% nano-SiO_2_ are shown in Figure 8. From the variation trend, it was observed that a certain amount of nano-SiO_2_ can improve the permeability resistance of concretes. With the dosage of nano-SiO_2_ increasing from 0% to 5%, the permeability resistance of concretes is improved significantly with increasing nano-particle dosage. The water permeation depth of concretes incorporating 5% nano-SiO_2_ is decreased by 55.6% compared to concrete without nano-SiO_2_. However, the water permeation depth of the concrete increases gradually with increasing nano-SiO_2_ content when the dosage of nano-SiO_2_ exceeds 5%, which indicates that the permeability resistance of the concrete starts to decrease at nano-SiO_2_ dosages greater than 5%. This suggests that the permeability resistance of the concrete starts to decrease when the nano-SiO_2_ content is greater than 5%.

The interspaces between cement particles can be filled by nano-SiO_2_ with small particle size, which significantly improves the density of concrete and block the water permeability channels. The characteristic of the interfacial transition zone (ITZ) between the aggregate and cement paste has a significant impact on the permeability resistance of concrete composites. Large amounts of nano-SiO_2_ will significantly improve structural characteristics of the ITZ in concrete composites [44]. Ardalan reported that the permeability resistance of concretes specimens was significantly enhanced by either spraying a layer of nano-SiO_2_ particles, or by being cured in a colloidal solution of water-diluted nano-SiO_2_ oxide [45]. The nano-SiO_2_ particle can be regarded as a type of catalyzer in accelerating cement hydration because of the large free energy of nano-particles and it also acts as kernels to reduce the crystal size of Ca(OH)_2_ [46]. As a result, the micropores and capillary pores on the surface and inside the concrete were filled.

Figure 9 shows the variation in water permeation depth of concrete specimens with increasing steel fiber dosage, on condition that the composite was combined with 5% nano-SiO_2_ and 15% fly ash. From the variations, it can be seen that the addition of steel fiber into concrete composite incorporating nano-SiO_2_ and fly ash considerably increases water permeation depth.

The elastic modulus of steel fibers is significantly greater than that of concrete; therefore, the addition of steel fibers would increase the tensile properties of concretes. This would effectively restrict the formation and development of initial cracks, and the porosity of the concrete could be reduced, which would be beneficial to improving the permeability resistance of concrete [47]. In addition, the interface with a certain amount of the cement paste wrapping both steel fibers and the aggregate is weaker than when the interface with the cement paste only wraps the aggregate. Therefore, tiny cracks would form more easily, and the water permeability would significantly increase. Compared to traditional high performance concrete, the incorporation of steel fibers has a significant impact on reducing the permeability of high performance concretes reinforced by steel fibers or steel fiber reinforced ultra-high performance concrete [48].

### 3.3. Freeze-Thaw Resistance

The relative dynamic elastic moduli of concrete incorporating nano-SiO_2_ with different nano-particle dosage is shown in Figure 10. It can be seen that concrete containing 3% nano-SiO_2_ exhibits the highest relative dynamic elastic modulus. After 25 freeze-thaw cycles, the relative dynamic elastic modulus of concrete containing 3% nano-SiO_2_ increased by 4.8% compared to concrete containing no nano-SiO_2_. When the nano-SiO_2_ content is greater than 3%, the relative dynamic elastic moduli increase with increasing nano-SiO_2_ content. The concrete containing 9% nano-SiO_2_ has the lowest relative dynamic elastic modulus.

Freeze-thaw resistance is one of the most significant evaluated durability properties of concrete structures in cold environments. The freeze-thaw damage to concrete is a result of small internal cracks that are full of water that freezes in each freeze-thaw cycle, and the greater volume of ice enlarge the small cracks. The strength of concrete structures after a certain number of freeze-thaw cycles is dependent primarily on the cement paste structure, including the micropore type, pore distribution, micropore size, number of capillaries, and porosity factor [49]. After a certain dosage of nano-SiO_2_ is added to the concrete, the increased density is beneficial to improving the freeze-thaw resistance of concretes [50]. As a result, the density of the cement paste can be significantly strengthened by the incorporation of an appropriate amount of nano-SiO_2_. However, the amount of internal defects in concrete could increase because of clustering of the nano-SiO_2_ particles if an excessive dosage is used, which would have an adverse effect on the freeze-thaw resistance of concrete.

Figure 11 shows the variations in the relative dynamic elastic moduli of the concrete incorporating 5% nano-SiO_2_ and 15% fly ash under various freeze-thaw cycles. As can be seen, the relative dynamic elastic moduli of each group are similar when the number of freeze-thaw cycles is less than 50. However, there is a clear decrease in the relative dynamic elastic modulus when the number of freeze-thaw cycles exceeds 50. With increasing steel fiber dosage, the relative dynamic elastic moduli of samples increased gradually. Compared to the concrete without steel fiber, the relative dynamic elastic modulus of concrete reinforced by 2.5% steel fibers was improved by 59.2% after the specimens were subjected to 50 cycles. When the number of cycles reached 100, the specimens without steel fibers and the specimens incorporating 0.5% steel fibers were all destroyed.

Steel fibers have remarkable bonding properties with the concrete matrix. The presence of steel fibers in concrete delays and restricts the occurrence and development of cracks. The volume of the water in the cracks will expand when frozen, which increases the crack development. With increasing freeze-thaw cycles, the cement paste wrapping the coarse aggregate is desquamated, and the dense structure of concrete is destroyed. The appearance quality of the concrete reinforced with 2.5% steel fiber is significantly better than that of the concrete reinforced with 1% steel fiber.

### 3.4. Cracking Resistance

Figure 12 and Figure 13 show the influence of SiO_2_ nano-particles on the evaluation parameters in cracking resistance of concretes. From Figure 12, it can be seen that the total cracking area per unit area of the concrete surface first decreases, and then increases gradually with increasing SiO_2_ nano-particles dosage. In particular, there is a sharp decrease in the total cracking area when the SiO_2_ nano-particles dosage increases from 1% to 3%. With further increases from 3% to 7%, the total cracking area remains approximately unchanged. The minimum total cracking area was observed on concrete specimens incorporating 5% SiO_2_ nano-particles. Compared to concretes containing 5% nano-SiO_2_, the concrete containing 9% nano-SiO_2_ exhibited a smaller total cracking area, and the total cracking area increased by 71.8%. As can be seen in Figure 13, with the nano-SiO_2_ content increasing from 0% to 7%, the number of cracks decreased from 10 to 3, whereas the number of cracks increased to 5 with further increases in nano-SiO_2_ content. It can be concluded that 5% to 7% nano-SiO_2_ could be the optimal dosage to restrict shrinkage cracking of the concrete.

When hydration occurs, the hydration product exhibits a significantly smaller volume than the volume of each component before hydration, and chemical shrinkage in the concrete occurs. The incorporation of nano-SiO_2_ restricts shrinkage of hydration products to a certain extent and reduces the possibility of crack formation [51]. The nano-SiO_2_ particles can absorb part of the free water in the concrete and decrease the plastic shrinkage of concrete because of the large specific surface area of the nano-particles. However, excessive amounts of nano-SiO_2_ absorb a large amount of water, and water evaporation on surface of the concrete could also consume significant quantities of water. Therefore, the concrete shrinkage could be increased by excessive amounts of nano-SiO_2_.

Figure 14 and Figure 15 show the impact of steel fibers on the evaluation parameters for cracking resistance of concrete containing fly ash and nano-SiO_2_. From Figure 14, it can be seen that cracking area of the specimen decreases gradually with increasing steel fiber fraction. In addition, the number of cracks exhibits an obvious decrease when the fiber fraction increases from 1.5% to 2.0%. On the surface of the specimen, the cracking area decreases from 777 mm^2^ to 248.3 mm^2^, a decrease of 68.1%, when the dosage of steel fiber increases from 0% to 2.5%. As can be seen in Figure 15, compared to the concrete without steel fiber, the number of cracks in the concrete reinforced with 2.5% steel fiber decreases from 5 to 2.

The steel fibers in the concrete matrix are distributed randomly, which can support the aggregates and prevent the aggregates from sinking. The surface disintegration of the composite was reduced and the uniformity of the concrete could be improved. As a result, the fibers can effectively prevent the concrete cracking. In addition, steel fibers offer significant reinforcement on the early-stage hardened concrete, and the tensile strength of concrete can be enhanced because the concrete composite exhibits lower elastic moduli than steel fibers, which restricts the generation of cracks. The stress field inside the concrete is more continuous and uniform, and the stress concentration on the micro-crack tip can be weakened, because they can restrict further crack propagation [52]. The ductility and fatigue cracking resistance of concrete was improved. The incorporation of nano-SiO_2_ enhances the bonding performance between steel fibers and the concrete matrix and is beneficial for the fibers to play an anti-cracking role. Shen et al. reported that the cracking stress, ratio of cracking stress to tensile strength, and increasing steel fiber fraction resulted in greater concrete cracking resistance [53]. The early stage shrinkage cracking of ultra-high performance concrete slabs can be completely controlled by the incorporation of 2% steel fiber; however, shrinkage cracks were clearly observed on the surface of slabs of ultra-high performance concretes without steel fibers [54].

## 4. Conclusions

In this study, the influence of steel fibers and SiO_2_ nano-particles on the durability of concrete composite incorporating fly ash was investigated. Based on the results obtained, the following conclusions can be drawn:Application of nano-SiO_2_ particles can significantly improve the permeability resistance, cracking resistance, freeze-thaw resistance, and carbonation resistance of concrete on condition that the SiO_2_ nano-particle dosage is within a certain limit. By increasing nano-SiO_2_ content, the permeability, cracking, freeze-thaw, and carbonation resistance of concrete were first enhanced, but then decreased. An excessive number of nano-SiO_2_ particles could adversely affect the durability of concrete.Incorporation of the correct amount of steel fibers improved the carbonation resistance of the concrete containing nano-SiO_2_; however, an excessive fraction of steel fiber would reduce the carbonization resistance of the specimens. In addition, the addition of steel fiber reduced the permeability resistance of the composite reinforced by nano-SiO_2_. Reinforcement of steel fibers enhanced the freezing-thawing and cracking resistance of concrete incorporating nano-SiO_2_. By increasing steel fiber volume content, the freeze-thaw resistance of concrete incorporating nano-particles exhibited an increasing trend, and the cracking resistance of the concrete decreased gradually.

## Figures and Tables

**Figure 1 materials-12-02184-f001:**
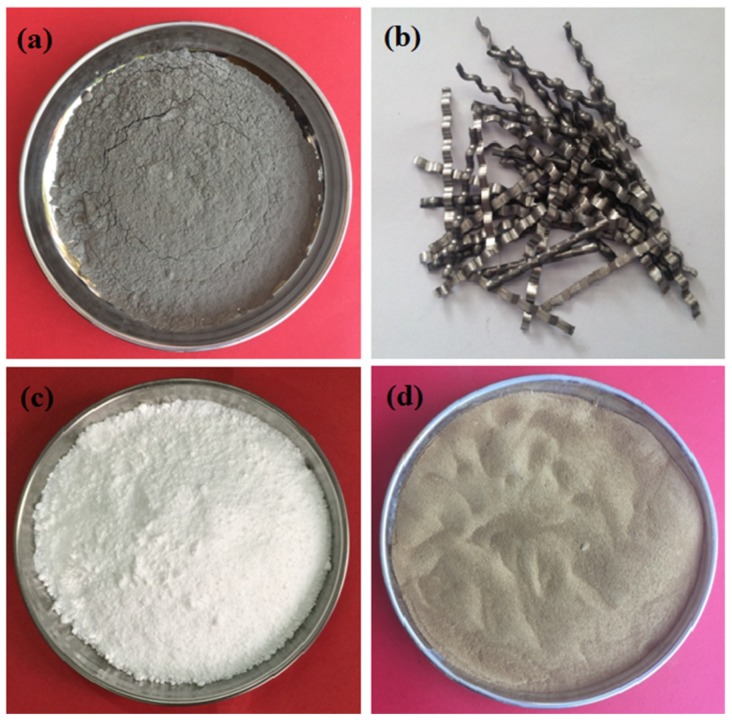
Raw materials used in study: (**a**) Class I fly ash, (**b**) Milling steel fiber, (**c**) Nano-SiO_2_, and (**d**) water reducing agent.

**Figure 2 materials-12-02184-f002:**
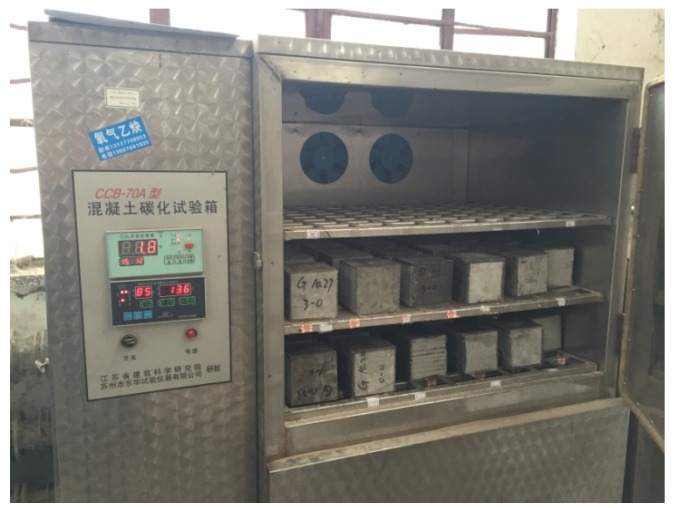
Carbonation test setup.

**Figure 3 materials-12-02184-f003:**
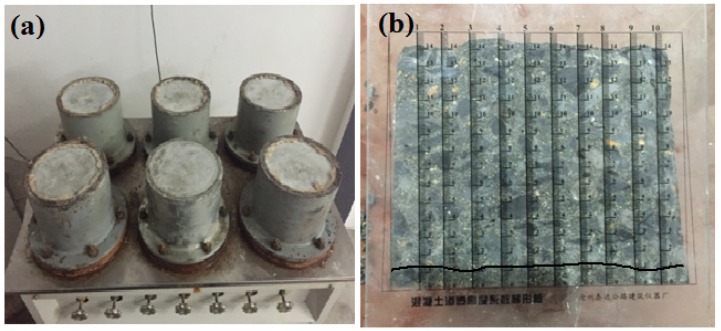
Anti-permeability test setup: (**a**) Anti-permeability machine, and (**b**) Measuring water permeability depth.

**Figure 4 materials-12-02184-f004:**
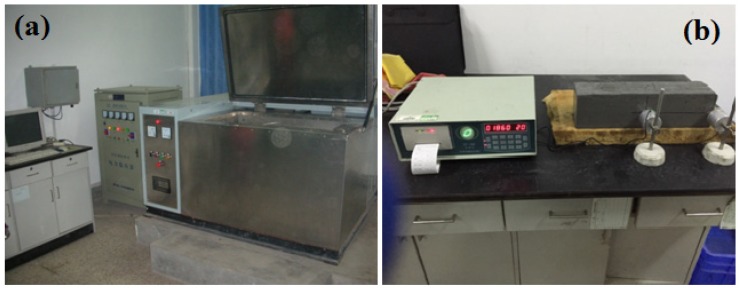
Freeze-thaw cycle test setup: (**a**) Freeze-thaw cycle test machine, (**b**) Measurement of dynamic modulus of elasticity.

**Figure 5 materials-12-02184-f005:**
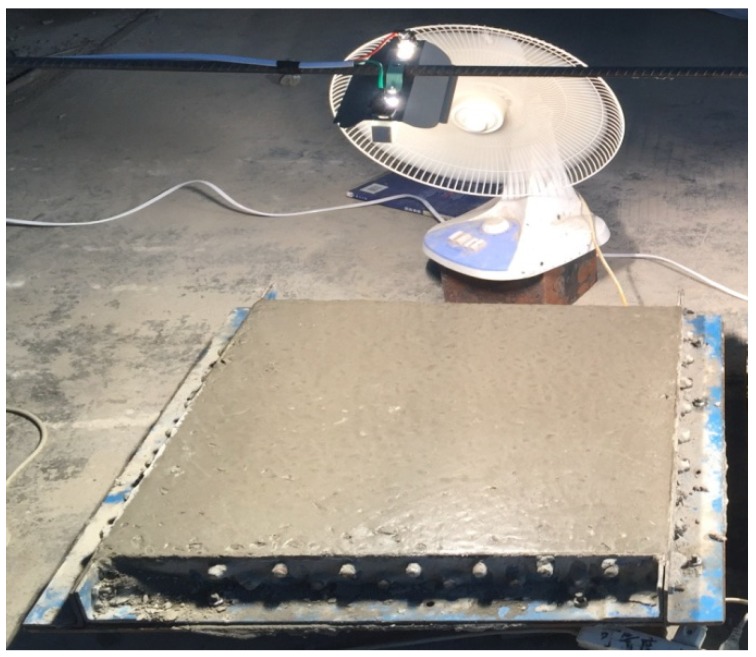
Cracking resistance test setup.

**Figure 6 materials-12-02184-f006:**
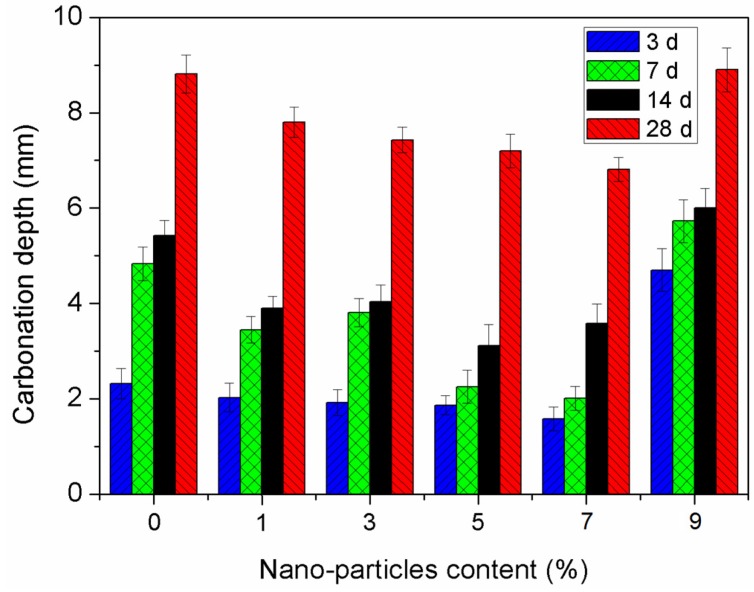
Effect of nano-particle content on carbonation depth.

**Figure 7 materials-12-02184-f007:**
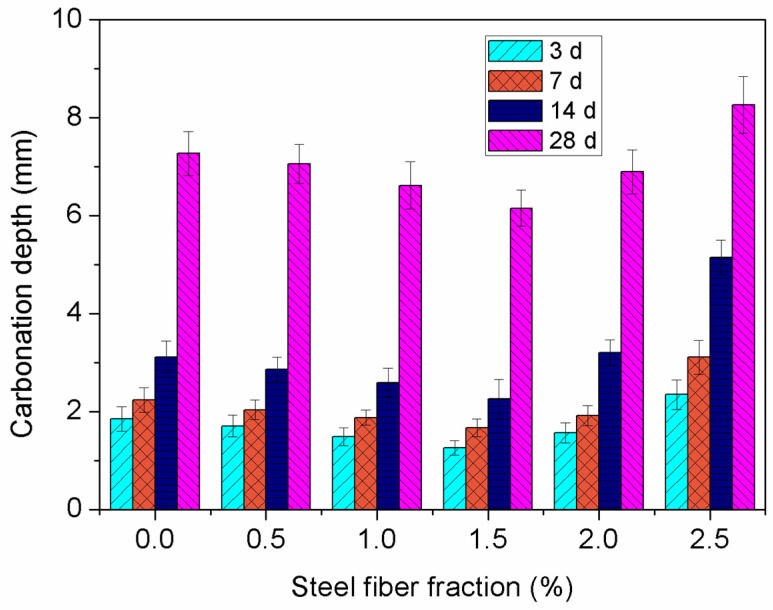
Effect of steel fiber fraction on carbonation depth.

**Figure 8 materials-12-02184-f008:**
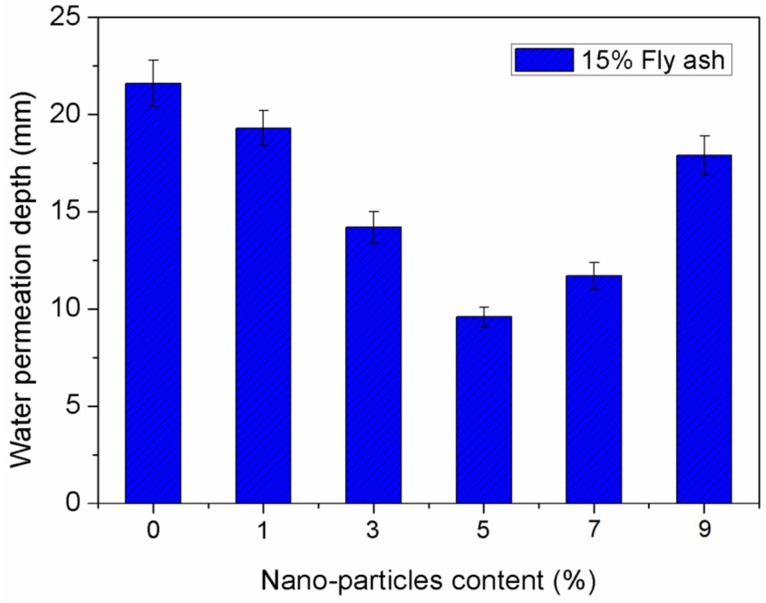
Effect of nano-particle content on length of water permeability.

**Figure 9 materials-12-02184-f009:**
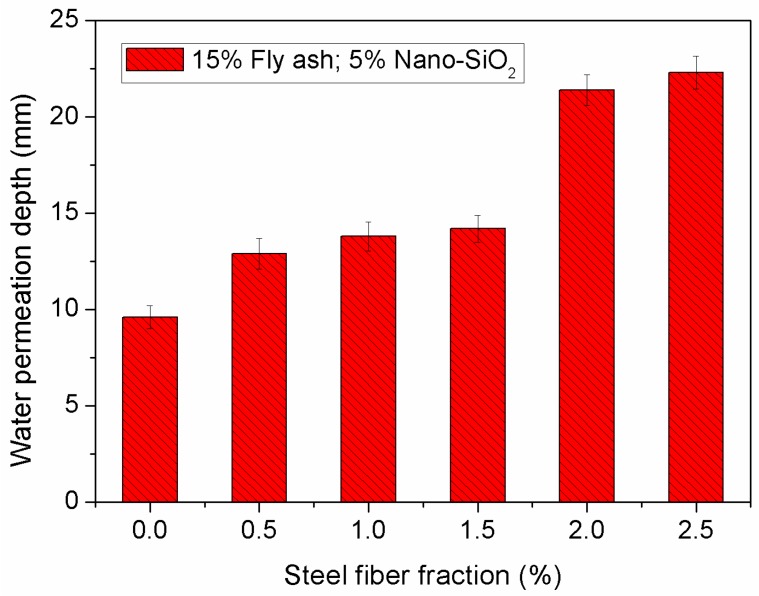
Effect of steel fiber fraction on length of water permeability.

**Figure 10 materials-12-02184-f010:**
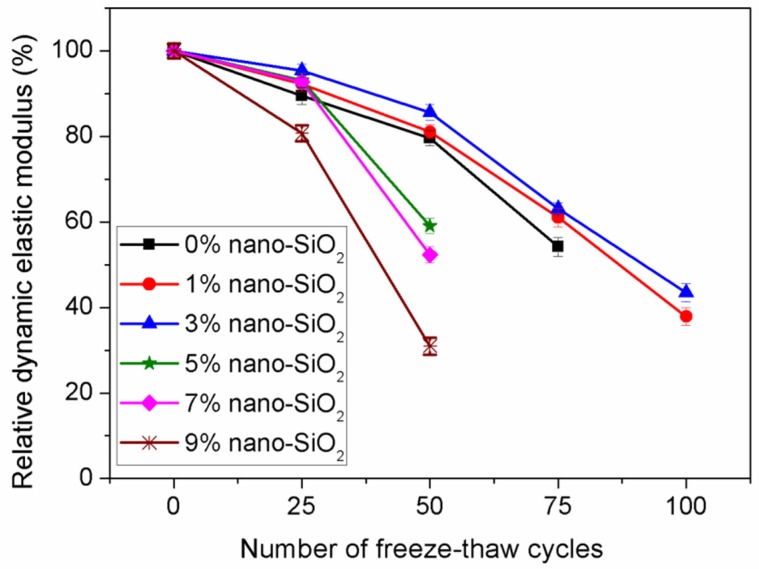
Effect of nano-particle content on relative dynamic elastic modulus.

**Figure 11 materials-12-02184-f011:**
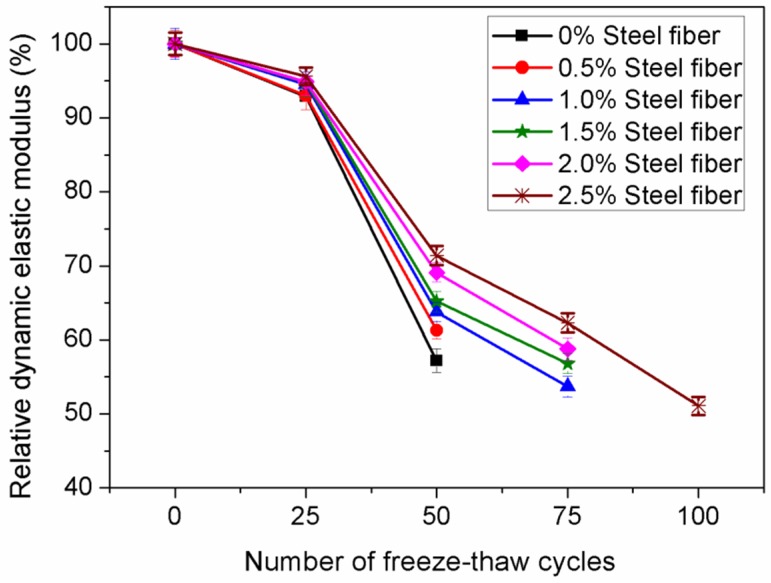
Effect of steel fiber fraction on relative dynamic elastic modulus.

**Figure 12 materials-12-02184-f012:**
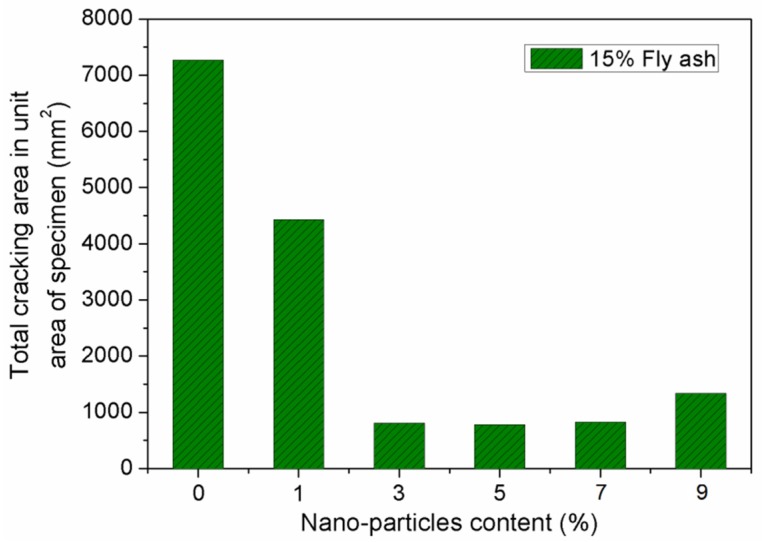
Effect of nano-particle content on total cracking area on per area of specimen.

**Figure 13 materials-12-02184-f013:**
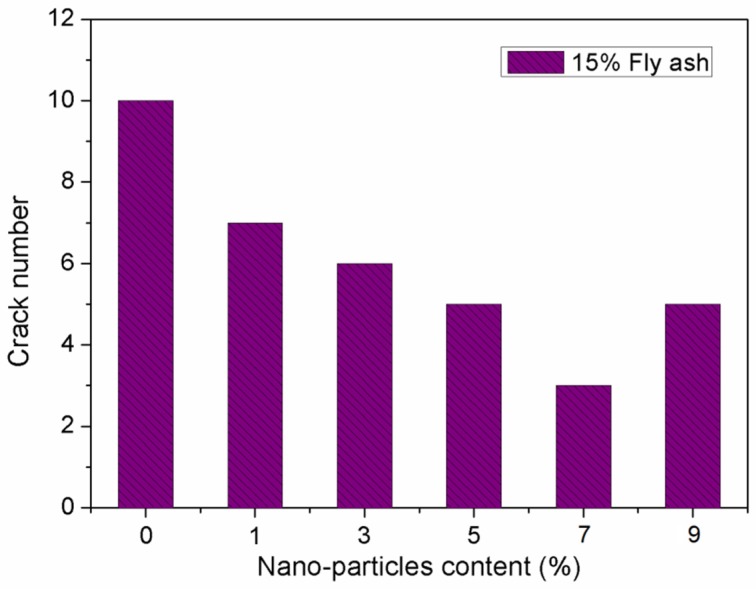
Effect of nano-particle content on crack number.

**Figure 14 materials-12-02184-f014:**
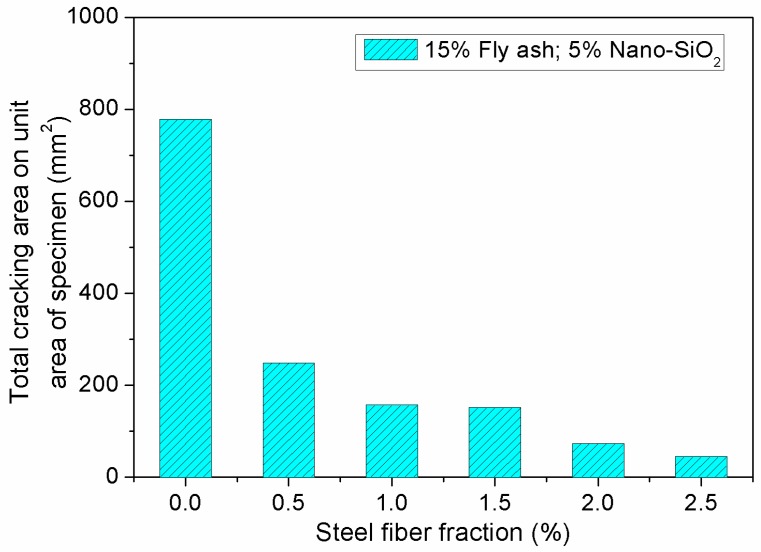
Effect of steel fiber fraction on total cracking area on unit area of specimen.

**Figure 15 materials-12-02184-f015:**
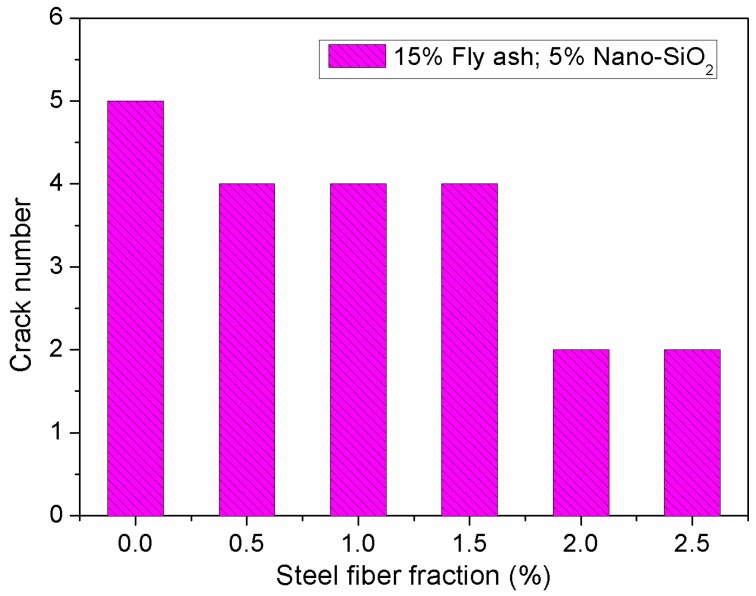
Effect of steel fiber fraction on crack number.

**Table 1 materials-12-02184-t001:** Properties of cement, fly ash, and steel fiber.

**Cement**	**Specific Gravity**	**Setting Time (min)**			**Compressive Strength (MPa)**		**Flexure Strength (MPa)**	
		Initial Setting		Final Setting	3d	28d	3d	28d
	3.16	90		286	26.1	53.8	5.37	8.52
Fly Ash	Moisture Content (%)	Ignition Loss (%)		SO_3_ (%)	Density (s)	Fineness (s)	Water Demand (%)	
	0.5	5.24		1.22	2.252	9.22	91	
Steel Fiber	Tensile Strength (MPa)		Length (mm)		Diameter (mm)		Length to Diameter Ratio	
	800		32		2.4		40	

**Table 2 materials-12-02184-t002:** Physical properties of nano-SiO_2_.

Average Particle Size (nm)	SiO_2_ Content (%)	Specific Surface Area (m^2^/g)	Bulk Density (g/cm^3^)	PH Value
30	99.5	200	0.055	6

**Table 3 materials-12-02184-t003:** Contents of binding materials and steel fiber.

Cement (kg/m^3^)	Fly Ash (kg/m^3^)	Nano SiO_2_ (%)	Steel Fiber (%)	Fine Aggregate (kg/m^3^)	Coarse Aggregate (kg/m^3^)	Water (kg/m^3^)	Water Reducing Admixture (kg/m^3^)
461.89	81.51	0	0	647	1151	158	5.98
456.46	81.51	1	0	647	1151	158	5.98
445.59	81.51	3	0	647	1151	158	5.98
434.72	81.51	5	0	647	1151	158	5.98
423.85	81.51	7	0	647	1151	158	5.98
412.98	81.51	9	0	647	1151	158	5.98
434.72	81.51	5	0.5	647	1151	158	5.98
434.72	81.51	5	1.0	647	1151	158	5.98
434.72	81.51	5	1.5	647	1151	158	5.98
434.72	81.51	5	2.0	647	1151	158	5.98
434.72	81.51	5	2.5	647	1151	158	5.98

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
