# Peer review of "Durability of Steel Fiber-Reinforced Concrete Containing SiO2 Nano-Particles"

_materials, 2019, doi:10.3390/ma12132184_

Round 1

Reviewer 1 Report

This manuscript presents an experimental study on the effects of nano SiO2 and steel fiber on the durability of reinforced concrete. I have to appreciate the authors’ effort to perform a larger munber of experiments to investigate various concrete properties. Unfortunately, the paper was not well written and lack of deep interpretations. In fact, the main mechanism to improve durability of concrete when adding nano silicdioxide is the reaction of SiO2 with portlandite to form secondary C-S-H. The authors are suggested to explain the effect of SiO2 addition in a concise way, try to avoid repetition. The same to steel fiber, it helps to increase tensile strength, bonding and other related properties.  Furthermore, the authors failed to prove the novelty of this work compared to the published ones.

The manuscript is written in bad English, though still readable. Detailed remarks are given below:

·       Lines 4-9:  check affiliation

·       Line 68: possesses?? 

·       Lines 100-103: it depends on the application and severe conditions . Carbonation is not always considered as a degradation process (it’s even helpful for non-reinforced concrete, e.g. porosity reduction, transport properties decrease (Phung et al. 2015))

·       Lines 120-121: you have to explain why those dosages were used.   

·       Lines 126-130: you used both fly ash and nano SiO2 in the mix, which act similarly in in term of hydration, how can you separate the roles of fly ash and nana SiO2?

·       Line 134: The water-reducing ratio of this admixture was 14% -->  I do not understand

·       Figs. 1-4: it is not needed to show 4 figures for raw materials, you may combine in one.

·       Fig. 4: wrong

·       Line 184: drying at 60oC can induce crack on samples, why do you choose drying in 48h?

·       Lines 193-197: can you link permeation depth with hydraulic conductivity?

·       Fig. 8: I can not see the water front.

·       Again, it is not needed to show all the apparatus in each single figure.

·       The setup to measure cracking resistance is not reliable.

·       You did not show how you measured carbonation depth. Can you put error bars in Fig, 12?

·       Lines 253-254: this statement does not relevant

·       Lines 256-264: not well explained, do you have any evidence to prove that porosity will increase if adding more than 7% SiO2?

·       Fig. 13: error bars

·       Line 287: The incorporation of steel fiber can efficiently block the diffusion channel of CO2 --> how? It may increase the diffusion due to ITZ effect.

·       Lines 293 – 295: not consistent with the composition in table 5, in which the cement contents are similar.

·       Lines 310-311: explanation

·       Line 337: initial??

·       Lines 338-340: not correct

·       Why do you choose dynamic elastic modulus to evaluate the freeze-thaw resistance?

·        Lines 370-372: not clear

·       Lines 386-399: not relevant to put it here

·       Line 424: why uniformly distributed?

·       Lines 430-432: reducing water does not mean cracking formation

·       Line 445: do not understand

·       Conclusion 2: rewritten

References:

Phung, Quoc Tri, Norbert Maes, Diederik Jacques, et al. 2015. Effect of limestone fillers on microstructure and permeability due to carbonation of cement pastes under controlled CO2 pressure conditions. Construction and Building Materials 82 (0):376-390.

Author Response

♥ Dear reviewer 1:

Thank you for your comments on our manuscript.

(1) Extensive editing of English language and style required

Response: In the revised paper, we have revised concerning syntax and spelling errors and improved the English expression as possible as we can.

(2) The paper was not well written and lack of deep interpretations. In fact, the main mechanism to improve durability of concrete when adding nano silicdioxide is the reaction of SiO2 with portlandite to form secondary C-S-H. The authors are suggested to explain the effect of SiO2 addition in a concise way, try to avoid repetition. The same to steel fiber, it helps to increase tensile strength, bonding and other related properties.

Response: According to your suggestion, we have revised the main effect mechanism of nano-SiO2 and steel fiber in a concise way.

(3)  Furthermore, the authors failed to prove the novelty of this work compared to the published ones.

Response: At the last paragraph of Introduction, we have indicated the novelty of this work.

(4) Lines 4-9:  check affiliation

Response: We have checked the affiliations of all the authors. The affiliations are all correct.

(5) Line 68: possesses?? 

Response: The word “possesses” is used wrongly on Line 68. We have revised the expression.

(6) Lines 100-103: it depends on the application and severe conditions . Carbonation is not always considered as a degradation process (it’s even helpful for non-reinforced concrete, e.g. porosity reduction, transport properties decrease (Phung et al. 2015))

Response: We have revised the improper analysis on Lines 100-103.

(7) Lines 120-121: you have to explain why those dosages were used.

Response: These nano-SiO2 contents and steel fiber volume dosages were chosen referring to the common dosage range of nano-SiO2 and steel fiber used in concretes in other researcher’s study and our previous study. We have explained why those dosages were used on Lines 120-121.

(8) Lines 126-130: you used both fly ash and nano SiO2 in the mix, which act similarly in in term of hydration, how can you separate the roles of fly ash and nana SiO2?

Response: Both of fly ash and nano-SiO2 were used in the mixtures in our study. However, the fly ash was used in all the mixes with the same content. So we can explore the effect of nano-SiO2 on durability of concretes by changing the content of nano-SiO2.

(9) Line 134: The water-reducing ratio of this admixture was 14% -->  I do not understand

Response: According to Chinese Standard “Concrete admixtures” (GB 8076-2008), the water-reducing ratio of an admixture can be defined as WR, WR=(W0-W1)/W0, where W0 is the water dosage of the control concrete mixture, and W1 is the water dosage of the concrete mixture added certain content of admixture, which has the same flowability with the control concrete mixture. We have added the related illustration in the revised manuscript.

(10)  Figs. 1-4: it is not needed to show 4 figures for raw materials, you may combine in one.

Response: According to your suggestion, we have combined Figs. 1-4 to one figure.

(11) Fig. 4: wrong

Response: We have combined Figs. 1-4 to one figure.

(12) Line 184: drying at 60oC can induce crack on samples, why do you choose drying in 48h?

Response: The specimens were being dried using a drying oven with temperature of 60 oC for 48 h. The drying time of 48 h was chosen according to Chinese Standard “Standard test methods for fiber reinforced concrete” (CECS 13-2009).

(13) Lines 193-197: can you link permeation depth with hydraulic conductivity?

Response: Based on the permeability resistance test described on Lines 193-197, the hydraulic conductivity can’t be measured because the water permeation quantity can be obtained from the test.

(14) Fig. 8: I can not see the water front.

Response: In practice, the boundary of water permeability is not very clear. In order to easily make out the boundary of water permeability, the specimen can be placed in the air for a certain time after it was split into two halves. We have replaced this figure using another figure with a lineation of water permeability on the splitting surface in the revised manuscript.

(15) Again, it is not needed to show all the apparatus in each single figure.

Response: According to your suggestion, we have combined the figures of apparatus of anti-permeability test, and apparatus of freezing-thawing cycle test to one figure, respectively.

(16)  The setup to measure cracking resistance is not reliable.

Response: The setup to measure cracking resistance was chosen according to Chinese Standard “Test methods of cement and concrete for highway engineering” (JTJ E30-2005).

(17) You did not show how you measured carbonation depth. Can you put error bars in Fig, 12?

Response: We have presented the detailed information on measuring carbonation depth on Section2.3 “Carbonation resistance test” and we have put error bars in Fig. 12 in the revised manuscript.

(18)  Lines 253-254: this statement does not relevant

Response: We have removed the statement on Lines 253-254.

(19) Lines 256-264: not well explained, do you have any evidence to prove that porosity will increase if adding more than 7% SiO2?

Response: We have revised the related explanation on Lines 256-264 in the revised manuscript.

(20) Fig. 13: error bars

Response: We have added the error bars in Fig. 13.

(21) Line 287: The incorporation of steel fiber can efficiently block the diffusion channel of CO2 --> how? It may increase the diffusion due to ITZ effect.

Response: According to the results of reference [39], we have added the related mechanism that steel fiber can efficiently block the diffusion channel of CO2 on Line 287.

(22)  Lines 293 – 295: not consistent with the composition in table 5, in which the cement contents are similar.

Response: The statements on Line 293-295 was just referred to the related analysis of reference [40]. For all the mixes in table 5, the water-binder ration keeps unchanged. As a result, we presented the related statements on Line 293-295.

(23)  Lines 310-311: explanation

Response: We have added the related explanation on Lines 310-311.

(24)  Line 337: initial??

Response: The expression of “initial” should be “initial cracks” on Line 310-311. We have revised the expression in the revised manuscript.

(25) Lines 338-340: not correct

Response: According to your suggestion, we have removed the wrong explanation on Lines 338-340.

(26) Why do you choose dynamic elastic modulus to evaluate the freeze-thaw resistance?

Response: According to the evaluation method in Chinese Standard “Standard for test methods of long-term performance and durability of ordinary concrete” (GB/T 50082-2009), we chose dynamic elastic modulus to evaluate the freeze-thaw resistance.

(27) Lines 370-372: not clear

Response: We have revised the mechanism analysis on Lines 370-372.

(28) Lines 386-399: not relevant to put it here

Response: According to your suggestion, we have removed the irrelevant statements on Lines 386-399.

(29) Line 424: why uniformly distributed?

Response: With the appropriate mixing technology and process, the nano-particles can be uniformly distributed in the concrete mixture.

(30) Lines 430-432: reducing water does not mean cracking formation

Response: We have revised the mechanism analysis on Lines 430-432.

(31)  Line 445: do not understand

Response: We have removed the improper explanation on Line 445.

(32) Conclusion 2: rewritten

Response: Conclusion 2 has been rewritten in the revised manuscript.

(33) References: Phung, Quoc Tri, Norbert Maes, Diederik Jacques, et al. 2015. Effect of limestone fillers on microstructure and permeability due to carbonation of cement pastes under controlled CO2 pressure conditions. Construction and Building Materials 82 (0):376-390.

Response: We have cited the suggested reference in the revised manuscript.

At last, thank you for your arduous work and instructive advice.

Reviewer 2 Report

The authors report the study of concrete samples in presence of different contents of nano-SiO2 particles and steel fibers. They report that the addition of nano-silica improves the permeability resistance, cracking resistance, carbonation resistance, and freezing-thawing resistance of concretes, when using proper amounts of SiO2. At the same time, proper amount of steel fibers can improve the carbonization resistance, enhance the freezing-thawing and cracking resistance.

The experimental results here reported might be interesting, but they are not clearly presented and an explanation was not thoroughly done. The introduction is long and dispersive. Reporting what nano-materials are, is really not necessary, while is unclear the actual novelty of this paper. As also reported by the authors, Zhang et al.[13] already studied the effect of steel fibers on concrete composites containing nano-silica, while Ismael et al.[23] already investigated the effects of SiO2 nano-particles on concrete prepared in presence of steel. You need to clarify this point.

The experimental analysis is not exhaustive; sometimes the discussion is superficial and some results are left unexplained. Overall, the paper is not well presented, and not clearly written.

In addition to the major concerns detailed above, I report below some specific comments:

·      According to the journal guidelines, authors should provide a concise and precise description of the materials and methods, which should be described with sufficient details to allow others to replicate and build on published results. Here the details about the experiments performed by the authors are missing.

·      The weight contents of nano-SiO2 are reported in percentages: is this weight percentage calculated with respect to cement?

·      You report the use of a fluidificant to reduce the water to cement ratio, which admixture did you use?

·      Tables 1, 2 and 3 could be merged for the sake of clarity.

·      Figures 1-11 are not necessary. There is no needing for showing pictures of commercial materials.

·      In section 2.3 you should describe the carbonation apparatus, not only the evaluation of the specimens after carbonation; in section 2.4 you should describe the details of the test to be used to evaluate the water permeability; n section 2.5 you should describe the method used to age the specimens by freeze-thaw and the method to evaluate its effect on the different formulations. Again, in section 2.6 the test is not clearly presented and the proper references should be cited when reporting the way of calculating the cracking resistance.

·      In figures 12-14 the error bars are missing.

·      At line 262 you report that the assembling of nano-SiO2 increase the porosity of concrete, it sounds confusing.

·      At line 273, you say that 5% of SiO2 was used, why exactly 5% and not 7%, since from fig 12 it results that the formulation containing 7% of silica would be better? If you decided to use 5% of silica because of the results arising from the permeability test, you need to reconsider the discussion section, since it is confusing as it is presented.

·      Lines 370-372: this explanation is not justified by the discussion here presented.

·      According to what stated at lines 399-400 and 428-430, in presence of more SiO2, more water should be consumed by the pozzolanic reaction. How do you explain that when more than 3% of silica is present the freeze-thaw resistance is worst?

·      Lines 444-461 are written in a different style.

Author Response

Dear reviewer 2:

Thank you for your comments on our manuscript.

(1) Extensive editing of English language and style required

Response: In the revised paper, we have revised concerning syntax and spelling errors and improved the English expression as possible as we can.

(2) The introduction is long and dispersive. Reporting what nano-materials are, is really not necessary, while is unclear the actual novelty of this paper.

Response: According to your suggestion, we have rewritten the introduction section of this paper, and removed the definition on nano-materials.

(3) As also reported by the authors, Zhang et al.[13] already studied the effect of steel fibers on concrete composites containing nano-silica, while Ismael et al.[23] already investigated the effects of SiO2nano-particles on concrete prepared in presence of steel. You need to clarify this point.

Response: Zhang et al. [13] and Ismael et al.[23] studied the effect of SiO2 nano-particles on mechanical properties, fracture properties and bonding performance of steel fiber reinforced concrete. However, very few literatures can be found on systemic investigation on the influence of nano-SiO2 on durability of concrete composite reinforced by steel fibers. The novelty of this paper is to reveal the influence of nano-SiO2 and steel fibers on the durability of the concretes composite. We have clarified this point in the revised manuscript.

(4) According to the journal guidelines, authors should provide a concise and precise description of the materials and methods, which should be described with sufficient details to allow others to replicate and build on published results. Here the details about the experiments performed by the authors are missing.

Response: We have added the related detailed information of the materials and methods in the revised manuscript.

(5) The weight contents of nano-SiO2 are reported in percentages: is this weight percentage calculated with respect to cement?

Response: The weight percentage of nano-SiO2 is calculated with respect to the total weight of cement and fly ash, and we have clarified this point in the revised manuscript.

(6) You report the use of a fluidificant to reduce the water to cement ratio, which admixture did you use?

Response: We used polycarboxylate-water reducing admixture to reduce the water to cement ratio and adjust the workability of cementitious composites. We have added the detailed information of this admixture in the revised manuscript.

(7) Tables 1, 2 and 3 could be merged for the sake of clarity.

Response: According to your suggestion, we have merged Tables 1, 2 and 3 as one table.

(8) Figures 1-11 are not necessary. There is no needing for showing pictures of commercial materials.

Response: Showing the pictures of fly ash, nano-particles and water reducing agent may be not necessary. However, I think show the shape of steel fibers used in this study may be helpful for replicating this study. Therefore, we combined the pictures of raw materials as one picture in the revised manuscript.

(9) In section 2.3 you should describe the carbonation apparatus, not only the evaluation of the specimens after carbonation; in section 2.4 you should describe the details of the test to be used to evaluate the water permeability; n section 2.5 you should describe the method used to age the specimens by freeze-thaw and the method to evaluate its effect on the different formulations. Again, in section 2.6 the test is not clearly presented and the proper references should be cited when reporting the way of calculating the cracking resistance.

Response: In section 2.3, we added the description on the carbonation apparatus.

(10) In section 2.4 you should describe the details of the test to be used to evaluate the water permeability;

Response: In section 2.4, we added the description on the details of the test to be used to evaluate the water permeability.

(11) In section 2.5 you should describe the method used to age the specimens by freeze-thaw and the method to evaluate its effect on the different formulations.

Response: In section 2.5, we added the description on the carbonation apparatus.

(12) Again, in section 2.6 the test is not clearly presented and the proper references should be cited when reporting the way of calculating the cracking resistance.

Response: In section 2.6, we added the description on the cracking resistance test and the proper reference was cited.

(13)  In figures 12-14 the error bars are missing.

Response: According to your suggestion, we have added the error bars in figures 12-15.

(14) At line 262 you report that the assembling of nano-SiO2 increase the porosity of concrete, it sounds confusing.

Response: At line 262, the explanation that the assembling of nano-SiO2 increase the porosity of concrete is not appropriate, and we have removed this statement.

(15)  At line 273, you say that 5% of SiO2 was used, why exactly 5% and not 7%, since from fig 12 it results that the formulation containing 7% of silica would be better? If you decided to use 5% of silica because of the results arising from the permeability test, you need to reconsider the discussion section, since it is confusing as it is presented.

Response: At line 273, 5% of SiO2 was used because of the results arising from the other tests of our previous study. It is known from our previous study that 5% nano-SiO2 addition has best reinforcement on mechanical properties and fracture performance on concrete composites [8, 13]. As a result, in the Mixes 7-11, the content of nano-SiO2 was kept unchanged with 5%. We have clarified this point in Section 2.1 in the revised manuscript.

(16) Lines 370-372: this explanation is not justified by the discussion here presented.

Response: We have revised the explanation at Lines 370-372 in the revised manuscript.

(17) According to what stated at lines 399-400 and 428-430, in presence of more SiO2, more water should be consumed by the pozzolanic reaction. How do you explain that when more than 3% of silica is present the freeze-thaw resistance is worst?

Response: Too large amount of nano-SiO2 will absorb more water. However, all the absorbed water is not consumed by the pozzolanic reaction because just only a part of nano-SiO2 will take part in pozzolanic reaction. The internal defects of concrete may increase because of the assembling of nano-SiO2 particles caused by overlarge dosage of nano-particles. As a result, the freezing-thawing resistance of concrete will be worse when overlarge content of nano-SiO2 was used.

(18) Lines 444-461 are written in a different style.

Response: There are a certain results on resistance of cracking for traditional concretes containing no nano-particles. So we can have our results compared with the current results. Maybe this section has another writing style.

At last, thank you for your arduous work and instructive advice.

Reviewer 3 Report

The topic of the paper is interesting and up-to-date, as the use of nanomodifiers in building materials is still one of the top fields of research. I see, however, the need of improvement of the paper. First, part of the illustrative material should be replaced; there is no need to present the typical (and not so impressive) images of the samples or conventional measuring equipment. Instead, the SEM images of microstructure of the composites are missing. They could support and help to explain the presented results. Second, the comprehensive review of using nanomaterials in concrete (including nanosilica in various forms) has been very recently published in Materials (E. Horszczaruk, Properties of Cement-Based Composites Modified with Magnetite Nanoparticles: A Review, Materials 2019,12(2), 326; https://doi.org/10.3390/ma12020326). This paper should be at least mentioned in bibliography. With the above amendments, I recommend to publish the reviewed paper.

Author Response

♥ Dear reviewer 3:

Thank you for your comments on our manuscript.

(1) Moderate English changes required

Response: In the revised paper, we have revised concerning syntax and spelling errors and improved the English expression as possible as we can.

(2) First, part of the illustrative material should be replaced.

Response: According to your suggestion, part of the illustrative material was replaced in the revised manuscript.

(3) There is no need to present the typical (and not so impressive) images of the samples or conventional measuring equipment. Instead, the SEM images of microstructure of the composites are missing. They could support and help to explain the presented results.

Response: We have removed the figure of samples in the revised manuscript. According to the suggestions of other reviewers, we have merged the pictures of the testing equipments in the revised manuscript. The SEM images of microstructures of the composites are very helpful to explain the presented results in our study. Thank you very much for your suggestion. We didn’t reserve extra samples for SEM measuring when we designed the experiments because there is no SEM equipment in our institute. However, we would add the experiments of SEM in our future study.

(4) Second, the comprehensive review of using nanomaterials in concrete (including nanosilica in various forms) has been very recently published in Materials (E. Horszczaruk, Properties of Cement-Based Composites Modified with Magnetite Nanoparticles: A Review, Materials 2019,12(2), 326; https://doi.org/10.3390/ma12020326). This paper should be at least mentioned in bibliography.

Response: According to your suggestion, we have citied the review article you mentioned, which was published in the journal of Materials.

 At last, thank you for your arduous work and instructive advice.

Round 2

Reviewer 1 Report

The revised manuscript has been significantly improved. Scientific content are OK now. However, it is still not good enough to be published. Two major issues need to be done before any consideration for publication:

Scientific language should be improved

Link between different durability properties should be discussed.

Few minor issues should be addressed as follows:

Line 23: check grammar

Lines 25-26: should use ‘’carbonation’’ though the text  

Line 68: revised

Line 115: determine

Lines 120-123: should not be there, would be in section 2.1

Lines 132-137: it is not scientific language, please revise

Table 2: check subscript

Fig. 1d: are you sure that was reducing water agent?

Carbonation test: why CO2 = 20%?

Lines 320-323: does not link to what you discuss later, should revise. You may get useful information on water permeability by following this works [1, 2].

Lines 336-336: not clear, C-S-H is not a small particle.

References:

[1] Phung QT, Maes N, De Schutter G, Jacques D, Ye G. Determination of water permeability of cementitious materials using a controlled constant flow method. Constr Build Mater. 2013;47(0):1488-1496.

[2] Scherer GW, Valenza JJ, Simmons G. New methods to measure liquid permeability in porous materials. Cement and Concrete Research. 2007;37(3):386-397.

Author Response

♥ Dear reviewer:

Thank you for your comments on our manuscript.

(1) Scientific language should be improved.

Response: In the revised paper, we have improved the scientific language as possible as we can.

(2) Link between different durability properties should be discussed.

Response: At the last paragraph of Introduction, we have added the discussion on link between different durability properties in the revised manuscript.

(3) Line 23: check grammar.

Response: We have revised the grammatical mistake on Line 23.

(4) Lines 25-26: should use “carbonation’’ through the text

Response: We have replaced the word “carbonization” with “carbonation” on Line 25-26.

(5) Line 68: revised

Response: We have revised the sentence on Line 68.

(6) Line 115: determine

Response: We have replaced the word “determin” with “determine” on Line 115.

(7) Lines 120-123: should not be there, would be in section 2.1

Response: We have removed the contents on Lines 120-123 to section 2.1 in the revised manuscript.

(8) Lines 132-137: it is not scientific language, please revise

Response: We have revised the section on Lines 132-137.

(9) Table 2: check subscript

Response: We have revised the related unit in Table 2.

(10)  Fig. 1d: are you sure that was reducing water agent?

Response: Yes Fig. 1d is the picture of reducing water agent, which is in powder.

(11) Carbonation test: why CO2 = 20%?

Response: According to the standard carbonation resistance test in reference [36], the concentration of CO2 should be 20%.

(12) Lines 320-323: does not link to what you discuss later, should revise. You may get useful information on water permeability by following this works [1, 2].

Response: We have revised the irrelevant discussion on Lines 320-323 and cited the two papers you suggested.

(13) Lines 336-336: not clear, C-S-H is not a small particle.

Response: We have revised this sentence in the revised manuscript.

At last, thank you for your arduous work and instructive advice.

Reviewer 2 Report

The authors have extensively revised the manuscript and now the novelty of the article is clear, the discussion has been improved and the methods have been clearly presented. Nevertheless, some work stil needs to be done. 

In general, the authors should be as concise as possible and many repetition are present and should be avoided (e.g. lines 15-24. 85-88, 115-119). 

How is it possible that when increasing the amount of nano-silica from 0 to 5 wt%, and without changing the amount of water reducing admixture, you used the same water to binder ratio? If you used the same amount of water in all samples, I am sure you obtained samples with very different workability and rheological properties... 

The cracking resistance test you present does not seem reliable and no references have been mentioned. If this test has been proved valuable, then you should cite the proper references. Otherwise, you need to prove that this test is correct and probably you need a comparison with a standard test to verify its reliability.  

In the results and discussion session, where do the error bard came from? Do they represent a standard deviation of many measurements? How many?

What you said at lines 267-269 it is wrong. Considering the error bars, it is now evident that there is no difference between samples containing 0 to 7 % of silica. Again, the decrease of carbonation depth showed in figure 7 is very little, when taking into account the error bars.

How can you say (lines 352-344) that the water permeation depth decrease when adding steel fibre up to 0.12%? 

Why no error bars where presented in figures 10 and 11? Did you only measure once one sample?

Here below some more comments:

Table 1: there is no point in labelling the samples if their acronymous is never used

line 66: nanometer effect?

lines 68-70: This is confusing. Actually nanoparticles consume plenty of water...

lines 70-82: what nanoparticles? It is mandatory to consider their composition, you can not imply address these improvements only to the dimension of the aggregates used in the previous studies

line 91-92: on the contrary, if you refer to the activity and specific area, these are not characteristics typical of nano-silica. A high surface area is present whenever small particles are considered.

line 175: there is no needs to discuss the presence of shelves

line 252: are you sure that you cited the proper reference?

Lines 432-433: actually the number of cracks does not change much until 2% of fibres are added

English and typos should also be checked again by the authors (e.g. line 34, 106 systemic, 129 applied, 130 .And, line 182 dried in a drying oven, line 183 twe, line 197 the specimen the specimen, line 226 10-20 and 2--4, check lines 237-241, at lines 272-283 it is hard to understand what you mean, review the sentence at line 312-315, line 503 2013, ).  

Author Response

Dear reviewer:

Thank you for your comments on our manuscript.

(1) In general, the authors should be as concise as possible and many repetition are present and should be avoided (e.g. lines 15-24. 85-88, 115-119).

Response: In the revised paper, we have revised the repetition on Lines 15-24, 85-88, 115-119.

(2) How is it possible that when increasing the amount of nano-silica from 0 to 5 wt%, and without changing the amount of water reducing admixture, you used the same water to binder ratio? If you used the same amount of water in all samples, I am sure you obtained samples with very different workability and rheological properties...

Response: In this study, we used the same water to binder ratio and water reducing admixture in order to reveal the effect of nano-silica content on workability of fresh concrete with the other mixing parameter unchanged. When we were preparing the samples, the workability of different mixes is different. However, the working performance for all the mixes is applicable to prepare the samples.

(3) The cracking resistance test you present does not seem reliable and no references have been mentioned. If this test has been proved valuable, then you should cite the proper references. Otherwise, you need to prove that this test is correct and probably you need a comparison with a standard test to verify its reliability.

Response: According to your suggestion, we have added the reference that we referred to in the section of 2.6 cracking resistance test.

(4) In the results and discussion session, where do the error bard came from? Do they represent a standard deviation of many measurements? How many?

Response: The error bar represents a standard deviation of three measurements.

(5) What you said at lines 267-269 it is wrong. Considering the error bars, it is now evident that there is no difference between samples containing 0 to 7 % of silica. Again, the decrease of carbonation depth showed in figure 7 is very little, when taking into account the error bars.

Response: As you indicated that it is evident that the difference between samples containing 0 to 7% of silica is not significant considering the error bars. However, it still can be seen from Figure 6 and Figure 7 that there is an obvious varying rule in the carbonation depth when the nano-particle content or steel fiber content is increasing.

(6) How can you say (lines 352-344) that the water permeation depth decrease when adding steel fibre up to 0.12%?

Response: The sentence on Lines 352-354 is wrong. We have deleted this stentence in the revised manuscript.

(7) Why no error bars where presented in figures 10 and 11? Did you only measure once one sample?

Response: According to your suggestion, we have added the error bars in figures 10 and 11.

(8) Table 1: there is no point in labelling the samples if their acronymous is never used.

Response: According to your suggestion, we have removed the column of Mix no in Table 3.

(9) line 66: nanometer effect?

Response: Compared with the traditional materials, nano particles have four unique nanometers effects including surface and interface effects, small size effect, quantum size effect, and macroscopic quantum tunneling effect.

(10) lines 68-70: This is confusing. Actually nanoparticles consume plenty of water...

Response: We have revised this sentence in the revised manuscript.

(11) lines 70-82: what nanoparticles? It is mandatory to consider their composition, you can not imply address these improvements only to the dimension of the aggregates used in the previous studies

Response: According to your suggestion, we have added the name of nano-particles on Lines 70-82.

(12) line 91-92: on the contrary, if you refer to the activity and specific area, these are not characteristics typical of nano-silica. A high surface area is present whenever small particles are considered.

Response: We have deleted the confusing expressions on lines 91-92 in the revised manuscript.

(13)  line 175: there is no needs to discuss the presence of shelves.

Response: According to your suggestion, we have deleted this sentence on line 175.

(14) line 252: are you sure that you cited the proper reference?

Response: We checked the reference carefully and we can sure that this reference is no problem.

(15)  Lines 432-433: actually the number of cracks does not change much until 2% of fibres are added

Response: We have revised the discussion on lines 432-433 in the revised manuscript.

(16) English and typos should also be checked again by the authors (e.g. line 34, 106 systemic, 129 applied, 130 .And, line 182 dried in a drying oven, line 183 twe, line 197 the specimen the specimen, line 226 10-20 and 2--4, check lines 237-241, at lines 272-283 it is hard to understand what you mean, review the sentence at line 312-315, line 503 2013, ). 

Response: We have revised the improper sentences and expressions you presented one by one in the revised manuscript.

At last, thank you for your arduous work and instructive advice.